# Phylogenomic Characterization of Lopma Virus and Praja Virus, Two Novel Rodent-Borne Arteriviruses

**DOI:** 10.3390/v13091842

**Published:** 2021-09-15

**Authors:** Bert Vanmechelen, Zafeiro Zisi, Sophie Gryseels, Joëlle Goüy de Bellocq, Bram Vrancken, Philippe Lemey, Piet Maes, Magda Bletsa

**Affiliations:** 1Laboratory of Clinical and Epidemiological Virology, Department of Microbiology, Immunology and Transplantation, Rega Institute for Medical Research, KU Leuven, Herestraat 49/Box 1040, 3000 Leuven, Belgium; bert.vanmechelen@kuleuven.be (B.V.); zafeiro.zisi@kuleuven.be (Z.Z.); bram.vrancken@kuleuven.be (B.V.); philippe.lemey@kuleuven.be (P.L.); piet.maes@kuleuven.be (P.M.); 2Evolutionary Ecology Group, Department of Biology, University of Antwerp, Universiteitsplein 1, 2610 Wilrijk, Belgium; sophie.gryseels@uantwerpen.be; 3OD Taxonomy and Phylogeny, Royal Institute of Natural Sciences, Vautierstreet 29, 1000 Brussels, Belgium; 4Institute of Vertebrate Biology, The Czech Academy of Sciences, Květná 8, 603 65 Brno, Czech Republic; joellegouy@gmail.com; 5Department of Zoology and Fisheries, Faculty of Agrobiology, Food and Natural Resources, Czech University of Life Sciences Prague, Kamýcká 129, 165 21 Prague, Czech Republic

**Keywords:** *Arteriviridae*, rodent-borne arteriviruses, host spectrum, cross-species transmission, virus evolution

## Abstract

Recent years have witnessed the discovery of several new viruses belonging to the family *Arteriviridae*, expanding the known diversity and host range of this group of complex RNA viruses. Although the pathological relevance of these new viruses is not always clear, several well-studied members of the family *Arteriviridae* are known to be important animal pathogens. Here, we report the complete genome sequences of four new arterivirus variants, belonging to two putative novel species. These new arteriviruses were discovered in African rodents and were given the names Lopma virus and Praja virus. Their genomes follow the characteristic genome organization of all known arteriviruses, even though they are only distantly related to currently known rodent-borne arteriviruses. Phylogenetic analysis shows that Lopma virus clusters in the subfamily *Variarterivirinae*, while Praja virus clusters near members of the subfamily *Heroarterivirinae*: the yet undescribed forest pouched giant rat arterivirus and hedgehog arterivirus 1. A co-divergence analysis of rodent-borne arteriviruses confirms that they share similar phylogenetic patterns with their hosts, with only very few cases of host shifting events throughout their evolutionary history. Overall, the genomes described here and their unique clustering with other arteriviruses further illustrate the existence of multiple rodent-borne arterivirus lineages, expanding our knowledge of the evolutionary origin of these viruses.

## 1. Introduction 

Over the past decade, the number of pathogens known to infect animals or humans has expanded considerably [1]. Major advances of high-throughput sequencing technologies along with the development of bioinformatics pipelines have greatly facilitated virus surveillance and discovery [2]. From large-scale surveillance efforts of wildlife, species-rich orders, such as bats and rodents, appear to be able to maintain and disseminate pathogens to a wide variety of other mammalian hosts [3]. Both rodent and bat hosts hold tremendous potential for zoonotic transmission and are considered important natural reservoirs of infectious diseases [4]. To understand the risk of virus spillover from wildlife to humans, it is important to elucidate the association between emerging infectious diseases and their zoonotic reservoirs and to identify host–pathogen interactions that could potentially threaten public health.

In the past few decades, several members of the family *Arteriviridae* have emerged as important animal pathogens, although there are currently no known arterivirus species capable of infecting humans [5]. Of particular interest is porcine respiratory and reproductive syndrome virus (PRRSV-1/2), which causes respiratory distress in newborn piglets, as well as reproductive failure in sows. Equally important to the farming industry is equine arteritis virus (EAV), which causes clinically variable but potentially life-threatening equine viral arteritis. Both the aforementioned arteriviruses have a significant impact on livestock health, resulting in a worldwide economic burden on pig and horse agriculture [6,7]. Many other arteriviruses are also known to be pathogenic, although the severity of their disease can vary greatly depending on the virus species and host, ranging from mostly subclinical in the case of lactate dehydrogenase-elevating virus (LDV) in mice to a lethal neurological disease in the case of wobbly possum disease virus (WPDV) in brushtail possums [8,9,10]. Furthermore, several simian arteriviruses are capable of causing severe hemorrhagic fever with high case fatality rates in their respective hosts [11]. 

All members of the family *Arteriviridae* have a positive-sense, single-stranded RNA genome, varying from 12–18 kb in length [12]. The genome can be subdivided into two parts, with the 5′ end encoding the non-structural, regulatory proteins and the 3′ end encoding the structural components of the virion. The expression of the viral polyprotein occurs directly from the genomic viral RNA and is achieved by joining the two large open reading frames (ORF1a and -b) on the 5′ end of the genome through ribosomal frameshifting. This polyprotein is subsequently processed into several non-structural proteins, including the viral polymerase [13]. The polyadenylated 3′ end of the genome acts, through the generation of intermediary negative-sense, subgenomic RNAs, as a template for the transcription of subgenomic mRNAs that encode the structural virus proteins. These include an envelope protein (E), a matrix protein (M), a nucleocapsid (N) and several glycoproteins (GP2, −3, −4, −5 and −5a). In the genomes of simian arteriviruses, an additional 3–4 ORFs encoding glycoproteins are located just upstream of the ORF2a [14,15]. In addition, recent research has shown that these major ORFs may represent only a part of the expansive coding capacity of arteriviruses [16].

Following the latest reorganization of the family *Arteriviridae*, there are currently 23 recognized arterivirus species spread across 13 genera in 6 subfamilies [17]. Furthermore, there are several more recently discovered arteriviruses that have not yet been classified but for which a (near) complete genome sequence is available. While most of these yet unclassified viruses were found in rodents and cluster within the currently recognized subfamily *Variarterivirinae*, some others were found in reptiles and likely represent novel subfamilies within the family *Arteriviridae*, or even separate families [18,19,20]. 

In the present study, we describe the discovery and genome organization of two novel arteriviruses, Lopma virus and Praja virus, which were detected in rodents from Mozambique and Tanzania, respectively. We explore their evolutionary relationships with other members of the family *Arteriviridae*, characterize virus–host phylogenetic relationships and propose their taxonomic clustering into two separate, novel genera in the *Variarterivirinae* and *Heroarterivirinae* subfamilies, respectively. 

## 2. Materials and Methods

### 2.1. Sample Collection and Illumina Sequencing 

Rodents were captured in four different countries between 2010 and 2013, with the aim to study the taxonomy and molecular ecology of rodent populations in sub-Saharan Africa [21,22,23,24,25,26,27,28,29]. In brief, individuals were captured in multiple localities in the Democratic Republic of the Congo, Ethiopia, Mozambique and Tanzania using various types of traps. Spleen, kidney and other organs were excised and stored in RNAlater (Qiagen, Leiden, The Netherlands) at −20 °C or in ethanol at room temperature. 

We revisited this sample collection, as part of a larger initiative to investigate the evolutionary relationships of hepaciviruses in small African rodents [30]. To uncover and reconstruct the microbiota present in those rodents, a subset of 42 hepacivirus-positive specimens was selected for Illumina (Appendix A). Prior to sequencing, total RNA was purified from kidneys and spleens based on the protocol described in [30]. RNA extracts were measured with the RNA Quantifluor System (Promega, Leiden, The Netherlands) and their quality was assessed on a Bioanalyzer 2100 using an Agilent RNA 6000 Nano chip (Agilent Technologies, Leuven, Belgium). Upon quantitation, samples were subjected to a ribosomal RNA (rRNA) depletion step using the Ribo-Zero rRNA Removal Kit (Illumina) and sequencing libraries were prepared with the NEXTflex Rapid Illumina Directional RNA-Seq Library Prep Kit (PerkinElmer, Hamburg, Germany). Barcoded libraries were combined into six different pools and paired end sequenced on an Illumina NextSeq 500 (Illumina) at Viroscan3D (Lyon, France).

### 2.2. Genome Assembly

Illumina reads were adapter and quality trimmed and subsequently de novo assembled using the CLC Genomics Workbench (v10.0.1). Arteriviral contigs were identified by performing a tBLASTx search of all contigs against a local database containing all available arterivirus RefSeq genomes [31]. In total, five samples were found to contain arteriviral reads, with Lopma virus being present in three samples and Praja virus in two. In the case of Lopma virus, near-complete genome sequences (~13.7 kb) could be assembled from the Illumina data of two of the samples, while the data of Praja virus yielded only fragmented assemblies. Based on these assemblies, primer sets were designed to determine the two genome sequences of Praja virus by PCR and subsequent Sanger sequencing. PCRs were performed using the Qiagen OneStep RT-PCR kit (Qiagen, Leiden, The Netherlands) and the following cycling conditions: 30 min at 50 °C for reverse transcription, followed by 15 min at 95 °C and 40 cycles of 30 s at 94 °C, 30 s at 55 °C and 1 min at 72 °C. A final elongation step was done at 72 °C for 10 min. The resulting PCR products were purified using PureIT ExoZap PCR CleanUp (Ampliqon, Odense, Denmark) and sequenced by Macrogen Europe (Macrogen Europe, Amsterdam, The Netherlands). The resulting amplicon sequences were joined with the fragmented Illumina assembly using Seqman (v7.0.0). The 5′ and 3′ sequence ends of all sequences were determined using the Roche 5′/3′ race kit, 2nd generation (Hoffmann-La Roche, Basel, Switzerland), employing the Qiagen OneStep RT-PCR kit for all PCR steps. For 5′ RACE, C-tailing was used instead of A-tailing, and the oligo-dT primer was replaced by an equivalent oligo-dG primer. Sequencing of the resulting amplicons was performed as described above. A complete list of all primers used is given in Appendix A. Host species identification was confirmed by reconstructing the cytochrome b gene from the generated Illumina data. GenBank accession numbers corresponding to the rodent cytochrome b gene sequences are provided in Appendix A. 

### 2.3. Dataset Compilation and Phylogenetic Analysis

Amino acid sequences of the ORF1ab were retrieved from each (putative) arterivirus species for which a coding complete genome sequence was available in GenBank. In addition to their amino acid sequences, metadata such as their accession number, information on the host species, sampling location and collection date were also obtained. In this dataset, we added the ORF1ab amino acid sequences of our novel genomes (*n* = 4), thus resulting in a final collection of 38 sequences (Appendix A). To the rodent subset of this dataset, we added all complete ORF1ab amino acid sequences originating from rodent hosts resulting in a collection of 38 sequences that represented our enriched rodent arterivirus dataset (Appendix A). 

Sequence alignment was performed using MAFFT (v7.407), followed by manual curation using Aliview (v1.18.1) [32,33]. From this alignment, the most conserved regions, corresponding to nsp4 and nsp8–10, were excised and realigned using MAFFT. The resulting alignment was trimmed with trimAl v1.4.rev15 (gappyout setting), resulting in 1259 included positions. MEGA 7 was used to estimate pairwise distances by counting the number of observed differences between pairs of sequences. For the mammalian-wide alignment, we used BMGE as an additional data curation step to further remove ambiguously aligned regions [34]. Phylogenetic trees were inferred using the LG + F+I + G4 amino acid substitution model implemented in the IQ-TREE software (v1.6.7) [35]. After obtaining support values from 1000 bootstrap replicates, maximum likelihood phylogenies were midpoint rooted and further annotated in FigTree (v1.4.3). Whole genome comparisons on nucleotide level were performed using PASC [36].

### 2.4. Co-Divergence Analysis 

To investigate virus–host co-divergence patterns, we plotted the topology of the host phylogeny opposite the virus tree and examined their degree of association. For this analysis, we revisited the rodent-wide dataset and enriched it with one additional ORF1ab amino acid sequence obtained from a hedgehog [37], as this was the only non-rodent derived virus clustering within the *Heroarterivirinae* subfamily. We constructed the host phylogeny based on a credible distribution of 1000 trees obtained from VertLife.org for all identified rodents and one hedgehog host species that harbor arteriviruses [38]. The co-phylo plot (or “tanglegram”) was estimated using the ape R package [39]. 

## 3. Results

### 3.1. Detection of Praja and Lopma Viruses 

A set of 42 hepacivirus-positive mixed tissue RNA extracts was selected for Illumina sequencing [30]. These samples were collected from four different sub-Saharan African countries (Democratic Republic of the Congo, Ethiopia, Mozambique and Tanzania) and represent 11 different rodent species (Table 1). The locations of the sampling sites in the four countries are shown in Appendix A. Upon assembly of the read data, five animals were found positive for the presence of arteriviruses: two *Praomys jacksoni* or Jackson’s soft-furred mice from Tanzania and three *Lophuromys machangui* or Machangu’s brush-furred rats from Mozambique. For two of the *Lophuromys* samples, a near-complete genome sequence was obtained in a single contig from the Illumina data, while the three other samples yielded only fragmented assemblies. The two complete genomes obtained from the *Lophuromys* individuals share ~93% nucleotide identity and appear to be variants of the same virus species. Based on the genetic identity (>90% over >80% of the genome), the virus identified in the third *Lophuromys* rat most likely belongs to that same virus species, although a complete sequence could not be obtained due to the limited resources available. The *Praomys* arteriviruses are likely genomic variants of the same virus species, with ~89% identity on the nucleotide level. For these samples, the fragmented Illumina assemblies were further completed by PCR and subsequent Sanger sequencing. RACE was used to complete the obtained genome sequences. The resulting genomes are 13,739/13,738 (Lophuromys 1/2) and 14,146 nucleotides (Praomys 1 and 2) long and are highly divergent from each other, sharing only 33–35% nucleotide similarity, as determined by PASC [36]. Based on the host in which they were discovered, these two novel groups of viruses were given the names: Lopma virus (*Lophuromys machangui*) and Praja virus (*Praomys jacksoni*), respectively. Both the Lopma virus and Praja virus are significantly different from all known arteriviruses, sharing respectively 47.3–47.7% (Lactate dehydrogenase-elevating virus 1) and 41.2–41.4% (forest pouched giant rat arterivirus) nucleotide similarity with their closest classified relatives. 

### 3.2. Genome Organization

Lopma and Praja viruses share a similar genome organization, following the archetypical genome layout that is shared by all currently classified members of the family *Arteriviridae*, with the exception of the subfamily *Simarterivirinae*. The genomes encode both the 1a and 1b parts of the viral polyprotein, as well as at least eight smaller ORFs found in all arterivirus genomes (2a, 2b, 3, 4, 5, 5a, 6 and 7) (Figure 1). These smaller ORFs encode the structural parts of the viroid: an envelope protein (E), the GP2, GP3, GP4, GP5 and GP5a glycoproteins, a matrix protein (M) and a nucleocapsid (N). To express the 1ab viral polyprotein, arteriviruses make use of ribosomal frameshifting [12]. When translating the 1ab mRNA, the ribosome will encounter a slippery sequence followed by an RNA secondary-structure-forming motif at the end of the 1a ORF. This combination of RNA motifs causes the ribosome to stall and potentially backtrack one position, resulting in a -1 frameshift. The slippery sequence is a heptanucleotide of the form XXXYYYZ, with X being any nucleotide, Y being an A or U and Z being an A, C or U. In arterivirus genomes, this motif typically takes the form of UUUAAAC, as is the case in the genomes of Lopma and Praja virus [12]. Here, this heptanucleotide is located at nucleotide positions 6354–6360 and 6303–6309/6304–6310, respectively, and is in both cases followed by RNA hairpin-forming sequences. 

Both viruses share a similar genome organization, having the 1a and 1b genes that together encode the viral polyprotein, as well as the 2a, 2b, 3, 4, 5, 5a, 6 and 7 ORFs encoding structural proteins. Both viruses also encode a transframe (TF) that is presumably expressed through the joining of this ORF with the 1a ORF via -2 ribosomal frameshifting. This TF is predicted to have a transmembrane organization, as shown in the corresponding probability plots. Plots were made using the TMHMM web server v2.0 (http://www.cbs.dtu.dk/services/TMHMM/, last accessed on 22 December 2020). 

In addition to -1 ribosomal frameshifting to generate the 1ab polyprotein, most arteriviruses are also known to employ -1/-2 frameshifting in the nonstructural protein 2 (nsp2) region of the 1a ORF, truncating the 1a polyprotein (-1) or joining it with a transmembrane region encoded on a different reading frame (-2) [40,41]. Nsp2 is one of the four arterivirus proteases encoded by the viral polyprotein, and similar to other papain-like cysteine proteases, its activity is dependent on the presence of a cysteine-histidine tandem that is conserved amongst all arteriviruses, including the here-described Lopma virus (C438-H451) and Praja virus (C538-H551) [42]. The nsp2 frameshifting occurs at a conserved GGUUUUU motif that can be found in the genome of all mammalian arteriviruses with the exception of EAV, although sometimes minor variations can be observed, especially in the case of simarteriviruses [40,41]. Located just downstream of the frameshift site is a secondary highly conserved motif, CCCANCUCC, that acts as a frameshift-stimulatory element (FSE). In the genome of Lopma virus, the frameshift site is located at nt 2502–2508, followed by the FSE (CCCAACUCC) 10 nucleotides downstream. The frameshift site can also be found in the genome of Praja virus, at nt 2512–2518/2513–2519, again followed by a FSE (CCCAGCUCC) 10 nucleotides downstream. 

### 3.3. Phylogenetic Analysis of the Mammalian-Borne Arteriviruses 

To determine the evolutionary relationships that Lopma and Praja viruses share with other members of the family *Arteriviridae*, we inferred the phylogeny of all arterivirus species based on the deduced ORF1ab amino acid sequences. None of the known arteriviruses have been found in more than one host species. By indicating the corresponding mammalian host species for each virus species, it becomes apparent that arteriviruses appear to form host-specific groups with a limited number of cross-species transmission events (Figure 2). In addition to the basally positioned shrew, equine and possum arteriviruses, which are the sole representatives of the subfamilies *Crocarterivirinae*, *Equarterivirinae* and *Zealarterivirinae*, respectively, three well-supported monophyletic clusters of viruses (clades A–C) can be distinguished. All simian arteriviruses group together in a monophyletic clade (clade B), which is synonymous with the subfamily *Simarterivirinae*. Contrary to the well-confined nature of arteriviruses in primate hosts, rodent-borne arteriviruses are interspersed throughout the phylogenetic tree and form multiple divergent lineages in clades A and C. Three distinct groups can be distinguished, two in clade A and one in clade C. All viruses in the first group in clade A were identified in rodents from the Muridae family, while for the other group, the origin was more heterogeneous, involving two rodent families (Cricetidae and Chinchillidae) and one pig family (Suidae). Together, these two groups form the subfamily *Variarterivirinae*. Lopma virus clusters within this subfamily as a sister lineage to the other Muridae-borne viruses. The rodent-borne arteriviruses in clade C have been discovered in Muridae and Nesomyidae rodents. In addition to the two strains of Praja virus, this last group consists of the non-rodent hedgehog arterivirus 1 (HhAV-1), which was discovered in an *Erinaceus europaeus* from the UK (MT415062), and the distantly related forest pouched giant rat arterivirus (KP026921), the only previously known rodent-borne arterivirus sampled in Africa. The clustering of Lopma virus and Praja virus as distinct lineages within their respective subfamilies was also validated by comparing a conserved part of the ORF1ab polyprotein for all known and putative arterivirus species (Appendix A). For both viruses, the similarity with the most closely related sequences is comparable to or even slightly below the values typically observed between members of the same subfamily, highlighting the divergent nature of these novel genomes (Appendix A). 

### 3.4. Co-Divergence of Rodent-Borne Arteriviruses 

As a next step in our analysis, we reconstructed the phylogeny of all full-length rodent-borne arteriviruses (Figure 3). Viruses originating from the same host species group together, except in the case of *Rattus exulans*, whose arteriviruses cluster in two separate, well-supported clades. With respect to the grouping by rodent family, it appears that a high degree of family-specific clustering can be identified, with only the Praja virus variants deviating from this pattern. Specifically, viruses obtained from rodents of the Muridae family form a large monophyletic clade including viruses hosted by rodents from the *Rattus*, *Bandicota*, *Mus* and *Lophuromys* genera and another monophyletic lineage that contains only strains from *Praomys jacksoni* rats (Praja virus). Additionally, arteriviruses found in Cricetidae rodents all group together, while an arterivirus hosted by a chinchilla (*Chinchilla lanigera*) falls outside of this clade as an outgroup. Finally, a single arterivirus obtained from a *Cricetomys emini* rat (family Nesomyidae) forms a sister lineage to the Praja virus variants, highlighting the relatively close phylogenetic relationship between the arteriviruses from these rodent hosts, even though their most recent common ancestor indicates a much deeper evolutionary branching event. 

To assess the co-divergence between arteriviruses and a subset of their hosts, we mapped the viral phylogeny to the evolutionary lineages of rodents and a single hedgehog host species (Figure 4). In general, we observe that arteriviruses share similar phylogenetic patterns with their hosts, with limited cases of cross-species transmission events. The majority of rodent-borne arteriviruses cluster together in lineages that are phylogenetically consistent with the lineages formed by their hosts. The most prominent examples of co-divergence are viruses and hosts from the *Rattus* and *Bandicota* genera, but also from the *Myodes*, *Caryomys*, *Microtus*, *Mus* and *Lophuromys* genera. However, some host shifting events are also evident in the tanglegram of Figure 4. Incongruous relationships were identified between the virus phylogeny and the *Cricetulus longicaudatus*, *Praomys jacksoni*, *Cricetomys emini*, *Chinchilla lanigera* rodent species as well as with the *Erinaceus europaeus* hedgehog. 

## 4. Discussion

In the last few years, arteriviruses have been discovered in a diverse group of mammals, the majority being in rodent and primate species. Despite the absence of human arterivirus reports, these pathogens pose a considerable threat to livestock health and wildlife [43]. In the present study, we characterized two new putative arterivirus species, Lopma virus and Praja virus, obtained from African rodents of the Muridae family. Our data increases the known diversity of African arteriviruses, previously limited to a single arterivirus genome recovered from an African pouched rat, by four more genomes and expands the species spectrum of arteriviruses by identifying two previously unsampled hosts.

A series of analyses were performed to characterize the genome organization of the Lopma and Praja viruses. Comparable to most arteriviruses, the genomes of both Lopma virus and Praja virus contain the necessary motifs required for ribosomal frameshifting. In addition to the expression of the 1ab polyprotein, ribosomal frameshifting is used by arteriviruses to join a transmembrane domain to the nsp2 region in the 1a ORF [40]. Nsp2 is a multidomain protein that acts as a subunit of the viral replicase and functions as a protease, possessing auto-cleavage activity [42,44]. It also acts as a co-factor for the nsp4 main protease and is known to play an important role in the regulation of the host humoral immune response [45,46]. The joining of a transmembrane region to nsp2 seems to be a conserved feature across multiple arterivirus species, but the function of the resulting ‘nsp2TF’ remains to be fully elucidated, although in the case of PRRSV it has been shown to act as a downregulator of the host’s innate immune response [45]. 

On a host species level, our data show that Lopma virus was detected in three *Lophuromys machangui* rodents from the same location in Mozambique. Brush-furred mice are found throughout sub-Saharan Africa, and little is known about their role as virus reservoirs [47]. However, recently we reported the discovery of two different paramyxovirus species within the same *Lophuromys machangui* individual, as well as several hepaciviruses in different *Lophuromys* rats [30,48]. Praja virus was detected in two *Praomys jacksoni* mice captured in Tanzania. Akin to brush-furred mice, Jackson’s soft-furred mice are found throughout East Africa, and besides our recent discovery of hepaciviruses in these mice [30], their role as virus hosts is currently unknown. Multiple strains of each virus were detected in multiple individuals of the same host species, making these rodents likely candidates to be (one of) the true reservoir species of Lopma virus and Praja virus, although either rodent might also be an accidental dead-end host. Targeted screening studies can further elucidate the host and geographical range of these novel arteriviruses and determine whether their infectivity is limited to specific rodent species, or if Lopma virus and Praja virus can also spread to other rodents and/or mammals. 

The relatively high diversity of arteriviruses in rodents and prominence of rodent hosts throughout the arterivirus tree seems to indicate a deep evolutionary history of arterivirus infections in rodent hosts. Although rodents are not currently believed to be active reservoirs for known livestock pathogenic arteriviruses, the results presented here do argue for rodents as ancient arterivirus reservoir hosts from where the virus has passed to other mammalian species [49]. This is in line with data from other virus families, where rodents are also believed to have acted as reservoir hosts aiding virus dissemination [22,30,50,51]. Lopma virus strains group together in the subfamily *Variarterivirinae* as sister lineages to several other yet unclassified rodent-borne arteriviruses. Most of these unclassified arteriviruses have been detected in rodents from the Muridae and Cricetidae families and show strong evidence of confinement to these hosts. Whether the propensity of such rodents to harbor arteriviruses indicates a possible evolutionary origin in these rodent families or if this is due to the sheer number of rodent species that these families contain needs to be interpreted with caution. Praja virus, conversely, does not fall within the subfamily *Variarterivirinae* but clusters distantly within the *Heroarterivirinae* subfamily. As described above, the closest relatives of Praja virus are HhAV-1 of the European hedgehog and forest pouched giant rat arterivirus, the latter being the only known rodent-borne arterivirus that lies outside of the subfamily *Variarterivirinae* [17,37]. This close phylogenetic relationship between HhAV-1 and the Praja virus strains could either be attributed to a relatively recent cross-species transmission event or long-term congruent evolution but elucidating this will first require additional sampling efforts of wildlife to uncover the complex virus evolutionary patterns behind arteriviruses infecting these mammalian orders and families. 

The taxonomy of the family *Arteriviridae* has been reorganized twice in recent years and there are currently no clear guidelines for the classification of novel arteriviruses. In 2016, it was decided to use the NCBI Pairwise Sequence Comparison (PASC) tool for taxon demarcation within the family *Arteriviridae*, using 39–41% and 71–77% as genus and species cut-offs, respectively [52]. According to these criteria, both Lopma virus and Praja virus should be classified as novel species. In 2018, however, the arterivirus taxonomy was reorganized, elevating the existing genera to subfamilies and several species to the genus or subgenus level, whilst also revising the rules for the nomenclature of arterivirus taxonomic groups [53]. This novel classification is based on comparing patristic pairwise distances in concatenated multiple sequence alignments of five conserved domains spread throughout the ORF1ab polyprotein: 3CLpro, NiRAN, RdRp, ZBD and HEL1. Because technical details on how this new classification was obtained have not yet been published, a formal analysis of the classification of Lopma virus and Praja virus is currently not possible. However, taking into account that the current arterivirus taxonomy is based on domain conservation in the ORF1ab polyprotein and given the clustering of Lopma and Praja virus based on phylogenetic analysis of their polyproteins, both viruses are likely to each represent a novel genus in the family *Arteriviridae*. This is further supported by the low amino acid similarity between Lopma virus and Praja virus and their corresponding most closely related sequence when comparing the regions of the ORF1ab polyprotein that contain the five abovementioned conserved domains. In accordance with the current ICTV nomenclature, these genera could be named *Xiarterivirus* and *Omicronarterivirus*, respectively.

In conclusion, we present here the complete genome sequences of four novel arterivirus strains that represent two new virus species. Both Lopma virus and Praja virus were discovered in African rodents and have the archetypical arterivirus genome organization. These sequences considerably increase the diversity of known arteriviruses from Africa, since only a single rodent arterivirus genome was previously available from the continent. Additionally, their unique positions in the arterivirus phylogenetic tree provide new insights into the evolution of rodent-borne arteriviruses, with Praja virus further illustrating the existence of rodent-borne arterivirus lineages outside the subfamily *Variarterivirinae*. Given the considerable impact arteriviruses have on agriculture and livestock health, further research is needed to identify the role that rodent hosts play in the spread of arteriviruses and to characterize the possible routes of transmission between and/or among those hosts.

## Figures and Tables

**Figure 1 viruses-13-01842-f001:**
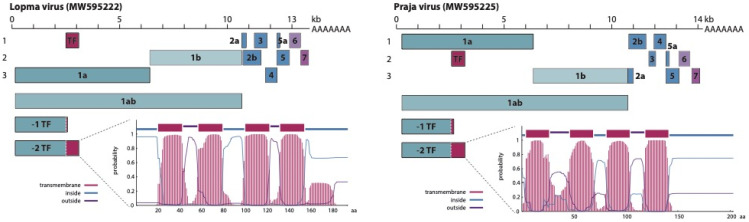
Genome organization of Lopma virus and Praja virus.

**Figure 2 viruses-13-01842-f002:**
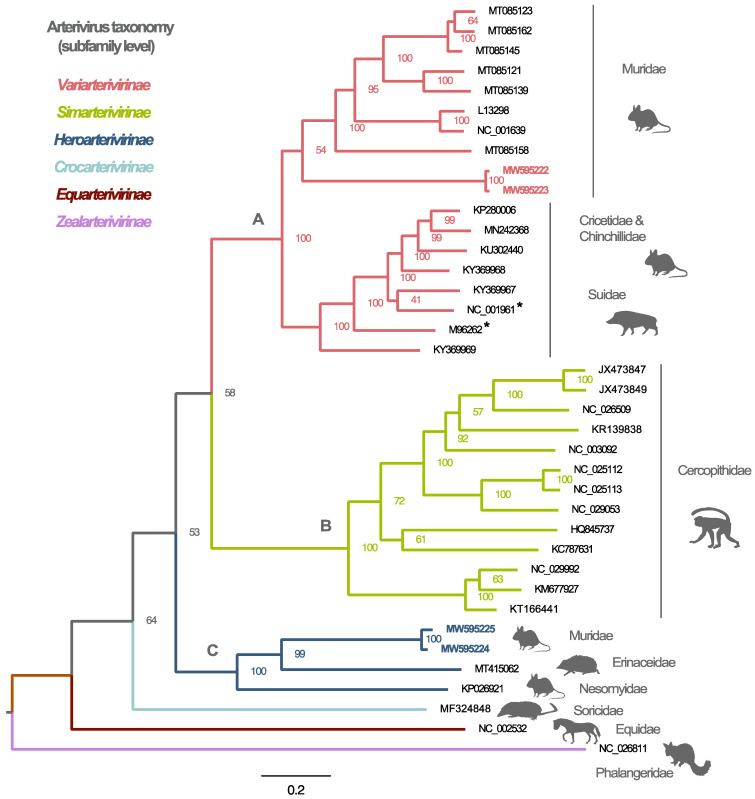
Phylogenetic reconstruction of the mammalian-borne arteriviruses. Maximum likelihood phylogenetic tree based on the ORF1ab of all putative arterivirus species for which coding-complete genomes are available. Silhouettes and descriptions in grey color denote mammalian host families. Numbers next to the internal nodes indicate bootstrap support, while branches are colored according to the assigned arterivirus subfamilies. Asterisks indicate the two porcine arterivirus genomes. Our novel genomes are labelled in highlighted bold accession numbers (red: Lopma and blue: Praja). Phylogenetic clustering shows that Lopma virus strains group within the subfamily *Variarterivirinae*, while Praja strains are mostly related to viruses from the subfamily *Heroarterivirinae*. Clades A, B and C have been provisionally named to facilitate discussion of the various mammalian arterivirus lineages.

**Figure 3 viruses-13-01842-f003:**
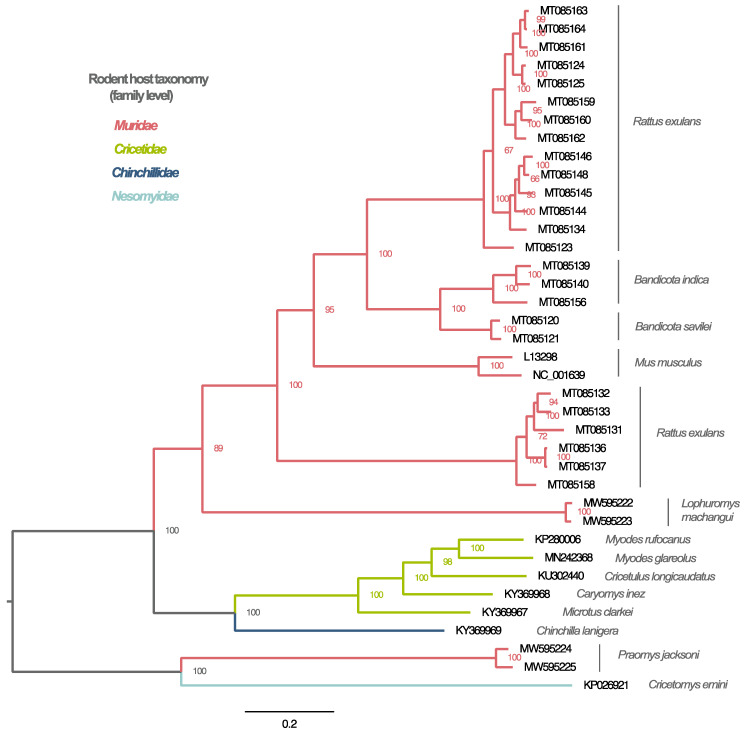
Phylogenetic reconstruction of the rodent-borne arteriviruses. Maximum likelihood phylogenetic tree based on the ORF1ab of all rodent arteriviruses for which coding-complete genomes are available. Numbers next to the internal nodes indicate bootstrap support, while branch colors denote taxonomically assigned rodent families. Descriptions in grey italics correspond to rodent species*. Phylogenetic clustering shows that rodent-borne arteriviruses group together at a rodent family level with only the Praja virus variants not following this pattern. (* footnote: For the rodent arteriviruses with GenBank accession numbers KY369968 and KY369967, the original GenBank submission file listed their hosts as “*Eothenomys inez*” and “*Neodon clarkei*”, respectively. To be consistent with the most up-to-date mammalian taxonomy, these species names were replaced by their homotypic synonyms, which are the “*Caryomys inez*” and the “*Microtus clarkei*”, respectively).

**Figure 4 viruses-13-01842-f004:**
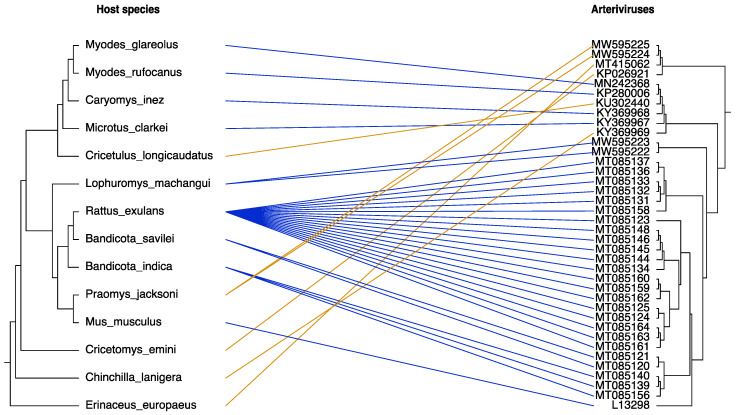
Tanglegram of arteriviruses and their rodent and hedgehog hosts. The host tree topology was inferred using the VertLife.org online tool (left phylogeny). The virus tree was constructed using the rodent-wide alignment with the addition of one hedgehog arterivirus sequence (right phylogeny). Interconnected lines represent the relationships between viruses and their hosts. Blue lines correspond to a phylogenetically congruent virus–host association, while orange lines highlight potential cross-species transmission events.

**Table 1 viruses-13-01842-t001:** Animals screened by Illumina sequencing.

Host Species	Positivity	Country of Origin
Acomys wilsoni	0/1	Tanzania
Graphiurus kelleni	0/1	Democratic Republic of the Congo
Lemniscomys striatus	0/1	Democratic Republic of the Congo
Lophuromys dudui	0/7	Democratic Republic of the Congo
Lophuromys laticeps	0/3	Tanzania
Lophuromys machangui	**3**/16	**Mozambique**, Tanzania
Lophuromys stanleyi	0/4	Tanzania
Mastomys natalensis	1/1	Tanzania
Micaelamys namaquensis	0/1	Mozambique
Praomys jacksoni	**2**/5	Democratic Republic of the Congo, **Tanzania**
Stenocephalemys albipes	0/2	Ethiopia

## Data Availability

The genomic sequences generated in this study were deposited in GenBank under accession numbers MW595222–MW595225.

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
