# Peer review of "Phylogenomic Characterization of Lopma Virus and Praja Virus, Two Novel Rodent-Borne Arteriviruses"

_viruses, 2021, doi:10.3390/v13091842_

Round 1

Reviewer 1 Report

I enjoyed reading the manuscript so much that I forgot I as reviewing it! So my congratulations to the authors for a well written and scientifically supported study.

My major concerns with the content are as follows.

  1. please provide information as a supplemental table on the GIS coordinates of where the animals that were selected were caught.
  2. Please provide information on how the species were identified. If molecular sequencing was used then please upload the gene to GenBank.
  3. Please include how the mammals were captured and how tissues were stored (in brief) so that readers don't have to rummage through 5 papers.
  4. Please provide the rational for why these animals were chosen
  5. Please provide a map on where the animals used were captured.

Reviewer 2 Report

The manuscript entitled “Phylogenomic characterization of Lopma virus and Praja virus, 2 two novel rodent-borne arteriviruses” is complete.

The authors discovered novel arteriviruses genome, including Lompa and Praja virus, and characterized the genome organization and phylogenetic relationship.

Overall, the experiments are carried out and described well, and all the results supports their conclusion. Moreover, it important to report the novel viruses, and viral genome in this field.
